# Physiological and Agronomical Response of Coffee to Different Nitrogen Forms with and without Water Stress

**DOI:** 10.3390/plants13101387

**Published:** 2024-05-16

**Authors:** Victor Hugo Ramirez-Builes, Jürgen Küsters, Ellen Thiele, Juan Carlos Lopez-Ruiz

**Affiliations:** 1Center for Plant Nutrition and Environmental Research Hanninghof, Yara International, 48249 Dülmen, Germany; 2Independent Researcher, Cra. 23D 69-43, Manizales 170008, Colombia

**Keywords:** coffee, nitrogen (N), ammonium form (NH_4_^+^), nitrate forms (NO_3_^−^), photosynthesis (Ps), chlorophyll, soil acidity, productivity

## Abstract

Nitrogen (N) is the most important nutrient in coffee, with a direct impact on productivity, quality, and sustainability. N uptake by the roots is dominated by ammonium (NH_4_^+^) and nitrates (NO_3_^−^), along with some organic forms at a lower proportion. From the perspective of mineral fertilizer, the most common N sources are urea, ammonium (AM), ammonium nitrates (AN), and nitrates; an appropriate understanding of the right balance between N forms in coffee nutrition would contribute to more sustainable coffee production through the better N management of this important crop. The aim of this research was to evaluate the influences of different NH_4_-N/NO_3_-N ratios in coffee from a physiological and agronomical perspective, and their interaction with soil water levels. Over a period of 5 years, three trials were conducted under controlled conditions in a greenhouse with different growing media (quartz sand) and organic soil, with and without water stress, while one trial was conducted under field conditions. N forms and water levels directly influence physiological responses in coffee, including photosynthesis (Ps), chlorophyll content, dry biomass accumulation (DW), nutrient uptake, and productivity. In all of the trials, the plants group in soils with N ratios of 50% NH_4_-N/50% NO_3_-N, and 25% NH_4_-N/75% NO_3_-N showed better responses to water stress, as well as a higher Ps, a higher chlorophyll content, a higher N and cation uptake, higher DW accumulation, and higher productivity. The soil pH was significantly influenced by the N forms: the higher the NO_3_^−^-N share, the lower the acidification level. The results allow us to conclude that the combination of 50% NH_4_-N/50% NO_3_-N and 25% NH_4_-N/75% NO_3_-N N forms in coffee improves the resistance capacity of the coffee to water stress, improves productivity, reduces the soil acidification level, and improves ion balance and nutrient uptake.

## 1. Introduction

The Coffea genus comprises 103 species [1], but all coffee market production and consumption is supported by two species, namely *C. arabica* L. (Arabica coffee) and *C. canephora* Pierre (Robusta Coffee). Arabica coffee accounts for 57.5% of the coffee trade, while Robusta comprises the remaining 42.5% [2]. In order of importance, the most important coffee-producer regions are South America, which contributes 48.1% of the total production, followed by the Asia–Pacific region with 29%, the Caribbean, Central America, and Mexico with 11.5%, and Africa with 11.3% [2]. Global coffee consumption increased from 90-million 60 kg coffee bags in 1990 to 167-million 60 kg green coffee bags in 2022 [2], representing a mean increase in the consumption rate of 2.4-million bags per year, with a discrete increase in the production areas, with reduction tendencies in some countries [3]. This situation clearly indicates that coffee productivity has increased systematically during recent years, mostly due to the introduction of agricultural intensification practices, including shade reduction, new coffee varieties, plant density, pruning-management practices, and mineral fertilization [4,5,6].

Worldwide coffee production is under threat due to climate change and variability [7,8]. The predicted increase in global mean air temperature of between 1.2 and 3 °C by 2050 will directly affect the climate sustainability of coffee-growing regions in the Americas, Asia, and Africa. The latest report by Grüter et al. [9] concluded that the main coffee-producing countries (Brazil, Vietnam, Indonesia, Colombia) are all seriously affected by climate change, with a strong decline in suitable areas depending on the scenarios—S (S1: 48–97% reduction; S2: 18–51% reduction). These results are even more dramatic than those previously reported by Bun et al. [7] and Ovalle et al. [8].

Regarding the impact of climate variability and change on coffee production, Phan et al. [10] conducted a review reporting that 20 of the 34 scientific publications indicated negative impacts and 14 reported mixed results. Harvest losses due to drought and climate variability were reported mostly in the Americas and could be as high as 70%. Most recently, Bilen et al. [11] reviewed 42 publications related to the same topic in coffee; of these, 35 indicated negative impacts on yield and production, four reported mixed results, and three revealed positive effects. The analysis conducted by Bilen et al. [11] identified an overall reduction in coffee yield across the three continents (America, Africa, and Asia). 

From a physiological perspective, climate change and variability provide advantages and disadvantages for coffee plants. The advantages include the benefit caused by the increase in atmospheric CO_2_ concentration, which leads to an increase in the photosynthetic capacity through an improvement in the diffusion of CO_2_ from the atmosphere to the chloroplasts, which simultaneously reduces the negative effect of photorespiration as more light energy is required under elevated CO_2_ concentrations [12]. The disadvantages include the fact that coffee is highly susceptible to drought stress, with significantly reduced photosynthetic capacity, growth, yield, and quality [13,14,15,16]. The magnitude of the damage caused depends on the intensity, duration, and physiological stage of the stress. The availability of N to plants determines their response to elevated CO_2_ concentrations [17] and drought stress more than any other environmental factor [18,19]. Proper N nutrition should be considered to allow for the better adaptation of coffee plantations to these new climate change scenarios [12,20].

Mineral fertilization is an important driver of productivity in intensive coffee-production systems [6,21,22]; for example, in Brazil, after 4 years without mineral N fertilization, the mean coffee yield declined by 5.3 times compared with the treatment at the optimum N rate [23]. In Colombia, the reduction in productivity varied according to the soil organic matter content from 50% to 80% after 3 years without N mineral application [24].

On the other hand, the application of fertilizers in coffee production, despite the high intensification of production systems, is still low. For example, in Brazil, coffee-production systems consume an average of 1.4-million tons of fertilizer per year [23]; this is divided among about 1,823,403 hectares of production [3], providing an average fertilization rate of 0.76 t ha^−1^. The optimal dose of N in coffee in Brazil is around 400 kg N ha^−1^ [25]; this would be equivalent to an optimal rate of mineral fertilization ranging between 1.2 and 1.5 t ha^−1^ year^−1^, depending on the nitrogen content of the fertilizer. In Colombia, the situation is similar; the optimal dose of nitrogen is around 300 kg ha^−1^ year^−1^, equivalent to a fertilization rate between 1.2 and 1.5 t ha^−1^.year^−1^, but the actual application rate is less than 0.5 t ha^−1^ year ^−1^ [26].

The lower consumption of mineral fertilizer is one of the reasons related to the low productivity of coffee-production systems, mainly in America and Africa, and becomes a limiting factor within the strategies of adaptation to climate variability and change. One of the reasons linked to the low level of mineral fertilization use among coffee farmers is the related costs: in intensified production systems, mineral fertilization is the second-most important cost after labor costs, and in mechanized production systems, it is the second most important cost after energy and mechanization [27].

In addition to the low application rates, the most commonly used mineral N source in coffee production is urea, which accounts for 50% of the total mineral fertilizer used, and urea is recommended in coffee production [23,26,28] due to its relative low cost and high N content, increasing the challenges posed for coffee productivity and sustainability due to the low efficiency generated by its high volatilization losses and large carbon footprint [29,30,31].

Nitrogen is the most important nutrient in coffee production during the growth stage and the second-most accumulated nutrient by coffee cherries during the productive stage [23,32]. N is the nutrient that contributes the most to coffee yield because it is a constituent of proteins, enzymes, coenzymes, nucleic acids, cytochromes, and caffeine [33,34,35]. Plants absorb most of their nitrogen through the roots in the forms of ammonium (NH_4_^+^) and nitrate (NO_3_^−^), and at a lower proportion as urea and/or organic forms, principally low-weight amino acids [35,36]. The preference for inorganic forms of nitrogen (NH_4_^+^ NO_3_^−^) varies among plants species and varieties, and it depends on environmental factors such as air and soil temperature, soil moisture, soil pH, and light [37,38,39,40,41]. In general, NO_3_^−^ may be more beneficial during droughts, as NH_4_^+^ is the primary source of N in flooded, freeze-damaged, and acidic soils [32]. Even though the metabolic cost of NH_4_^+^ assimilation is lower than that of NO_3_^−^ [38], most plants are sensitive to NH_4_^+^, and the long-term application of NH_4_^+^ usually inhibits plant growth and, in many cases, is toxic [42,43].

Plants fed with urea as the sole N source may also have the same toxic symptoms as those fed with NH_4_^+^, mainly when the nitrification is inhibited and higher amounts of NH_4_^+^ remain in the soil solution, reaching toxic concentrations [42,43]. On the other hand, plants use NO_3_^−^ as a preferred nitrogen source, and it acts as a signaling molecule in the various important physiological processes required for growth and development [44].

Considering that N is critical for coffee productivity, and that it acts as a key driver of climate change adaptation and mitigation [45], this study evaluated the physiological and agronomical responses of coffee plants to different forms of N under drought-stress conditions in controlled and field environments, with the aim being to guide farmers to make better decisions related to N management in coffee-production systems.

Under controlled conditions, three trials were conducted with the aim being to compare the influence of several nitrogen forms, NH_4_-N and NO_3_-N, and their combinations on the growth, nutrient uptake, photosynthesis (Ps), and chlorophyll content of coffee plants, as well as soil fertility using quartz sand and organic soil as the growth medium. Trial 1 compared the influence of the N forms on growth and nutrient uptake without drought stress. Trials 2 and 3 evaluated the influence of the interaction between N forms and water levels on growth, nutrient uptake, soil pH, Ps, and the chlorophyll content of the leaves using quartz sand and organic soil as the growth medium. The field trial evaluated, over the course of 4 years, the agronomical performance of four N forms with respect to the control without N.

## 2. Results

### 2.1. Effect of the Interaction between Nitrogen Forms and Water Level on Coffee Growth under Greenhouse Conditions

In all three greenhouse trials, nitrogen forms had a notable influence on coffee plant growth, which was expressed in dry matter accumulation (DW). In the first trial, without water stress using soil as the growth medium, the greatest growth occurred in treatments with 75% NH_4_-N/25% NO_3_-N and 50% NH_4_-N/50% NO_3_-N, with a total DW accumulation of 41.3 and 39.2 g plant^−1^, respectively, being significantly different from treatments with 100% NH_4_-N/0% NO_3_-N and 0% NH_4_-N/100% NO_3_-N, which accumulated a DW of 26.3 and 23.8 g plant^−1^, respectively (Table 1).

For the second and third trials, a significant reduction in the growth of coffee plants was observed due to the water deficit and a significant interaction between water level and nitrogen forms. In the case of Trial 2, as the nitrate proportion in the nutrient solution increased, the total dry biomass increased, reaching the highest total DW accumulation of 21.9 g plant^−1^ in the treatment with 100% nitrates (0% NH_4_-N/100% NO_3_-N) without water stress, as well as 16.4 g plant^−1^ in the treatment with water stress (Table 1, Figure 1).

In Trial 3, using soil as the growth medium, the interaction between nitrogen forms and water level was not statistically significant, with only a simple effect from the N form and water level. The lowest DW was observed with and without water stress treatment with 0% NH_4_-N/100% NO_3_-N, at 8.3 and 16.4 g plant^−1^, respectively, while the treatment with 75% NH_4_-N/25% NO_3_-N without water stress induced the highest DW accumulation with 47.1 g plant^−1^, and the treatment with 100% NH_4_-N/0% NO_3_-N with water stress led plants to accumulate a total DW of 26.5 g plant^−1^ (Table 1, Figure 1).

### 2.2. Effect of the Interaction between Nitrogen Forms and Soil Moisture on Nutrient Uptake

Regardless of the growth medium used (sand or soil), the nitrogen forms and their interaction with water levels showed significant effects on the uptake of nitrogen and cations by coffee plants (Table 2). In the first trial using soil as the growth medium without water stress, it was observed that the treatments 75% NH_4_-N/25% NO_3_-N and 50% NH_4_-N/50% NO_3_-N induced the uptake of the highest amounts of nitrogen, with a mean uptake of 1.236 and 1.165 g plant^−1^, respectively. Meanwhile, the plants under the 25% NH_4_-N/75% NO_3_-N treatment absorbed significantly more cations (K, Ca, and Mg), with a mean uptake of 1.089 g plant^−1^ for K^+^, 0.283 g plant^−1^ for Ca^2+^, and 0.081 g plant^−1^ for Mg^2+^.

In the second trial using sand as the growth medium, the interaction between nitrogen and the water level was statistically significant; water stress reduced the absorption of nitrogen and cations, but in the treatments with higher proportions of nitrogen in a nitric form like 50% NH_4_-N/50% NO_3_-N and 0% NH_4_-N/100% NO_3_-N, the absorption of nitrogen and cations such as K^+^, Ca^2+^, and Mg^2+^ was significantly improved (Table 2).

For the third trial, using soil as the growth medium, it was observed that the interaction between nitrogen forms and water level was only significant for the absorption of nitrogen and magnesium. Under conditions of adequate water supply or non-water stress, the highest nitrogen absorption occurred in treatments with 75% NH_4_-N/25% NO_3_-N as in Trial 1, with a mean N uptake of 0.827 g plant^−1^, while under conditions of water stress, the highest absorption occurred in treatments with a higher proportion of nitrogen in an ammonium form (100% NH_4_-N/0% NO_3_-N) with a mean N uptake of 0.807 g plant^−1^.

In the case of cation uptake, under conditions of water stress, plants under the 25% NH_4_-N/75% NO_3_-N treatment presented the highest absorption of calcium with a mean Ca^2+^ uptake of 0.197 g plant^−1^, and without water stress, the treatment with 50% NH_4_-N/50% NO_3_-N uptake includes the highest amount of Ca^2+^ with 0.329 g plant^−1^. 

Without water stress conditions, the treatment with 75% NH_4_-N/25% NO_3_-N induces the uptake of the highest amount of Mg^2+^, at 0.109 g plant^−1^, and under water-stress conditions, the highest K^+^ uptake was reached with the treatment 75% NH_4_-N/25% NO_3_-N at 0.071 g plant^−1^. The K^+^ uptake under both water levels was high in the treatments with 100% to 75% NH_4_-N/25% NO_3_-N, with a mean uptake of 0.579 and 0.574 g plant^−1^ without water stress and 0.867 and 0.894 g plant^−1^ under water-stress conditions (Table 2). 

### 2.3. Effect of the Interaction between Nitrogen Forms and Water Level on Soil pH and Nitrogen Contents

The soil pH significantly changes when different nitrogen forms and water levels are present. In Trials 2 and 3, where sand and soil were used as the growth medium, soil pH increased as the proportion of nitrogen in the NO_3_-N form in the nutritional solution increased. In Trial 2, where sand was used as the growth medium, without water stress, the pH of the soil increased at a rate of 0.024 units per percent of nitrogen in the NO_3_-N form, while in the water-stressed treatment, the soil pH showed a significant increase only in the treatments with 25% to 0% NH_4_-N/75 to 100% NO_3_-N (Figure 2A).

In Trial 3, where soil was used as the growth medium at both water levels, the pH increased linearly with the increase in nitrogen proportions in the NO_3_-N form, but it had at lower rates than those observed in Trial 2 in sand (Figure 2B), with increasing rates of 0.0169 and 0.0141 units per percent of nitrogen in the nitric form with and without water stress, respectively.

The differences in soil acidity with and without water stress, and the higher rate of acidification in the sand compared to the soil, can be explained by the NH_4_-N content in the growth medium. In the trial where sand was used as the growth medium (Trial 2), treatments with 100% to 75% NH_4_-N/0 to 25% NO_3_-N showed the highest NH_4_-N concentrations at 9.52 and 3.91 mg 100 g^−1^ without water stress and 11.12 and 5.51 mg 100 g^−1^ with water stress, respectively (Table 3).

In Trial 3, using soil as the growth medium, NH_4_-N concentrations were also significantly higher in treatments with high proportions of nitrogen in the ammonium form, such as treatments 100% to 75% NH_4_-N/0 to 25% NO_3_-N, but they were much lower in quartz sand conditions, with a mean concentration of 0.046 and 0.028 mg.100 g^−1^ without water stress and 0.274 and 0.352 mg.100 g^−1^ with water stress (Table 3). 

The NO_3_-N contents were significantly higher in treatments with 0% NH_4_-N/100% NO_3_-N, with a mean concentration of 0.30 mg.100 g^−1^ and 1.44 mg.100 g^−1^ in quartz sand without and with water stress, respectively, and were much higher in soil, with a mean value of 12.1 mg.100 g^−1^ and 19.18 mg.100 g^−^ without and with water stress, respectively (Table 3).

### 2.4. Effect of the Interaction between Nitrogen Forms and Water Level on Photosynthesis and Chlorophyll Contents

Water stress significantly reduced the photosynthesis rates in coffee in both trials where sand and soil were used as the growth media (Figure 3). The interaction between water level and nitrogen forms in both trials was significant. 

When coffee plants were grown in quartz sand, without water-stress conditions, the mean Ps was 6.63 μmol CO_2_ m^−2^ s^−1^ in the treatment with 25% NH_4_-N/75% NO_3_-N, while under conditions of water stress, the highest Ps values were observed in the treatments with 50 to 25% NH_4_-N and 50% to 75% NO_3_-N, being 3.55 and 3.97 μmol CO_2_ m^−2^ s^−1^, respectively. Treatments with 100% NH_4_-N/0% NO_3_-N with and without water stress provided the lowest Ps, with mean rates of 0.40 and 1.81 μmol CO_2_ m^−2^ s^−1^, respectively (Figure 3A).

When soil was used as the growth medium, without water stress, the treatment 75% NH_4_-N/25% NO_3_-N provided the highest Ps, with a mean rate of 5.80 μmol CO_2_ m^−2^ s^−1^. Under water-stress conditions, the treatment with 50% NH_4_-N/50% NO_3_-N provided the highest Ps, with a mean rate of 3.27 μmol CO_2_ m^−2^ s^−1^. Under these growing conditions, treatment with 0% NH_4_-N/100% NO_3_-N provided the lower Ps with and without water stress, with a mean rate of 1.21 and 1.69 μmol CO_2_ m^−2^ s^−1^, respectively (Figure 3B).

Water levels and nitrogen forms had a significant effect on the chlorophyll content in coffee (Figure 4). Water stress increased chlorophyll content in coffee, being significantly higher in the treatment with 50% NH_4_-N/50% NO_3_-N at 889.7 units of N-tester compared to 701 units in the same nitrogen treatment without water stress. The water-stress treatments with 0% NH_4_-N/100% NO_3_-N provided the lowest chlorophyll contents (694.6 N-tester units), while under the conditions of a good water supply, the lowest chlorophyll content was observed in plants subjected to the 100% NH_4_-N/0% NO_3_-N treatment at 648.8 N-tester units (Figure 4).

### 2.5. Influence of Nitrogen Forms on Coffee Productivity and Chlorophyll Contents at Field Level

During the first 2 years of coffee harvest, no significant differences were observed in productivity between the plants exposed to different N forms and the control without N. After the 3rd year of treatment, significant differences were observed in yield among plants exposed to different nitrogen forms (harvest 2020): the treatments exposed to 50% NH_4_-N/50% NO_3_-N, and 25% NH_4_-N/75% NO_3_-N displayed the highest productivity, which was significantly different with respect to the urea and 75% NH_4_-N/25% NO_3_-N treatments, and the values for all the nitrogen treatments were significantly different with respect to those for the control without nitrogen (Figure 5).

The mean coffee productivity after 4 years of the trial and three harvest seasons was significantly different among the N forms. The treatments with 50% NH_4_-N/50% NO_3_-N and 25% NH_4_-N/75% NO_3_-N displayed the highest mean productivity, with 14,568 and 15,979 kg ha^−1^ of coffee cherries, respectively, while the urea and 75% NH_4_-N/25% NO_3_-N treatments produced a yield of 12,151 and 12,082 kg ha^−1^ of coffee cherries, respectively. The yields for these treatments were significantly different from that of the control, producing 10,077 kg ha^−1^ of coffee cherries (Figure 5).

During the final year of harvest, the chlorophyll content in the leaves significantly changed over the course of the year and among treatments (Table 4). Treatment without nitrogen reduced the chlorophyll content, with a mean level of 810.1 N-tester units, and the treatments with 25% NH_4_-N/75% NO_3_-N provided the highest chlorophyll content, with a N-tester reading of 1023.0 (Table 4).

## 3. Discussion

### 3.1. Interaction of N Forms-Water Level on Physiological Response of the Coffee Plants

In this study, under controlled conditions, it was clearly observed that water stress significantly reduces biomass accumulation (DW), photosynthesis rate (Ps), chlorophyll content, and nutrient uptake. If the buffer capacity of the soil is low without the presence of organic carbon (Trial 2 in quartz sand), the higher the participation of NO_3_^−^, the better the physiological parameters, due to the fact that the treatments with 100% to 75% NH_4_-N showed the lowest DW and Ps (Figure 1A,B, Figure 2A and Figure 3A, Table 1 and Table 2), indicating a negative effect of NH_4_^+^ feeding due to the toxic accumulation of NH_4_^+^ interacting with the strong acidification of the growth medium (pH < 5.0), this being more critical under water stress.

When the growth medium was soil (Trials 1 and 3), treatments with higher proportions of nitrates (100% NO_3_-N and 75% NO_3_-N) showed the lowest DW and Ps (Figure 1C,D, Figure 3B and Figure 4, Table 1 and Table 2), mainly due to the higher accumulation of nitrates in the soil generated by the nutrient solution plus the soil nitrification (Table 3), with this being most critical under water-stress conditions. 

In the case of quartz sand, the low response displayed by the physiological parameters could be related to the low buffer capacity of the soil, significantly influencing the strong acidification in the treatment, with NH_4_-N accounting for between 100% and 50% of the total N (Figure 2A), meaning that acidification likely reduced the nitrification, as well as increasing the NH_4_-N concentrations in the growth medium for these treatments (Table 3). Meanwhile, in the trial using soil as the growth medium (Trial 3), the pH increases linearly with the NO_3_-N concentration, promoting nitrification in NH_4_-N-rich treatments and generating higher NO_3_-N concentration in 100% NO_3_-fed treatments (Table 3).

Several studies clearly indicate that, in NH_4_^+^-fed plants, some NH_4_^+^ is translocated from the roots to the shoots, but this usually accounts for a small proportion of the total nitrogen moving in the xylem. Some of the reduced nitrogen compounds may have originally been formed in the leaves and been transferred to the roots, but a considerable amount taken up by plants is assimilated in the roots in the form of amino acids like asparagine and glutamine [39], and coffee is no exception; according to Mazzafera and Gonzalves [33], 37.5% of the total N and 90% of the N in the amino acid fraction in coffee sap is present in the form of asparagine and glutamine. When nitrates are the nitrogen source, the proportion of nitrogenous compounds in the xylem represented by NO_3_^−^ ions is much higher, indicating that nitrates are primarily assimilated in the shoots of the plants, varying considerably between plant species [39,41]; in coffee, 51.9% of the total N transported in the sap is NO_3_^−^ [33].

When the highest content of nitrates is available in the soil, excessive NO_3_^−^ uptake reduces N assimilation and uptake. The response to excessive NO_3_^−^ feeding is species-dependent; for example, Duan et al. [43] found in blackberry plants that NO_3_^−^-fed plants were more likely to display the formation of reactive oxygen species (ROS) and malondialdehyde (MDA) than NH_4_^+^ (or urea)-fed plants, leading to oxidative stress, an imbalance between oxidants and antioxidants, and the inhibition of root cell division and elongation. Meanwhile, when NO_3_^−^ was used as a sole N source, the levels of ascorbic acid (AsA) and reduced glutathione (GSH) increased significantly, which may be because the roots produced a large number of free radicals with the NO_3_^−^-N treatment.

In terms of abiotic stress, Pissolato et al. [46] found that plants grown in a 100% NO_3_^−^ nutrient solution were more tolerant to water deficit, and this response was associated with increase nitric oxide (NO) production and high NR activity in roots, increasing the activity of antioxidant enzymes, photosynthesis, stomatal conductance, and root growth.

When plants are supplied with NO_3_^−^, PEP carboxylase activity in the leaves is high but low in plants supplied with NH_4_^+^; this difference is less obvious at pH 6.0 than at pH 4.0. In the roots, PEP carboxylase activity becomes greater in plants supplied with NH_4_^+^ at pH 4.0 than in plants supplied with NO_3_^−^ at the same pH, and the enzyme activity becomes higher in plants supplied with either ammonium or nitrates at pH 6.0. The PEP carboxylase fulfils an anaplerotic function, allowing for the synthesis of carbon skeletons that are required for amino acid synthesis during the assimilation of both nitrates and ammonium. The higher activity of PEP carboxylase in the roots of NH_4_^+^-grown plants reflects the need for more carbon skeletons to be available in the roots for amino acid synthesis when NH_4_^+^ is the nitrogen source.

When the nitrogen source is NO_3_^−^, the activity of the enzyme in the leaves should be higher, as the assimilation of nitrates occurs mostly in the shoots of the plants [39]. This was corroborated by Carr et al. [47], who found significantly lower amino acid contents in coffee leaves when these plants were feed with a nutrient solution containing 100% NO_3_^−^, while in the treatments with a balance between NO_3_^−^ and NH_4_^+^, the amino acid content in the leaves increased. Amino acids are important N compounds responsible for long-distance N distribution in plants.

The excessive NH_4_-N level in the soil observed in Trial 2, and the excessive NO_3_-N level observed in Trial 3 (Table 3), potentially interfered with the PEP carboxylase activity, NH_4_^+^ assimilation, and, finally, with the formation and translocation of amino acids and proteins. Carr et al. [47] found that coffee plants fed with 100% NH_4_-N or 100% NO_3_-N are less efficient in incorporating inorganic N into proteins compared with those fed with 50% to 12.5% NH_4_-N/50% to 87.5% NO_3_-N, suggesting that most of the inorganic N is accumulated as a soluble amino acid.

Nitrates (NO_3_^−^) are readily mobile in the xylem and can also be stored in the vacuoles of roots, shoots, and storage organs. For the N in NO_3_^−^ to be incorporated into organic structures, nitrates must be reduced to NH_4_^+^. Most of the ammonium, whether originating from nitrate reduction or from direct uptake from the soil solution, is normally incorporated into organic compounds through the roots, although some NH_4_^+^ may also be translocated to the shoots, even if the plants receive nitrate as the sole N form [35].

The reduction of nitrate to ammonium is mediated by two enzymes: nitrate reductase (NR), which catalyzes the two-electron reduction of nitrates to nitrites (NO_2_^−^), and nitrite reductase, which transforms nitrites to ammonium in a six-electron transfer process [35,48]. To prevent the accumulation of nitrites, which are toxic to plant cells, NR activity is regulated by several mechanisms, including enzyme synthesis, degradation, and reversible inactivation, as well as the regulation of effectors and the concentration of the substrate. The concentration of NR is increased by light, sucrose, and cytokinin, whereas glutamine, a primary product of N assimilation, represses the NR [35,49]. Carr et al. [47] found a lower abundance of glutamine in coffee leaves in plants fed with 100% NO_3_-N compared with those fed with 50% NH_4_-N/50%NO_3_-N. The low DW and Ps rates registered in Trial 3 could be associated with NR inactivation due to the high NO_3_^−^ content in the soil, being almost 10 times higher than for the same treatment in Trial 2 (Table 3). Meanwhile, the low physiological performance of the plants undergoing treatment with 100% NH_4_-N/0% NO_3_-N in Trials 1, 2, and 3 could be directly linked to a reduction in the NR activity due to the high NH_4_^+^ content in the soil solution, as was reported by Carr et al. [47] and Wang et al. [50].

Independently of the source of the NH_4_^+^ (nitrate reduction, photorespiration, lignin biosynthesis, N_2_ fixation in legumes, or senescence inducing N remobilization) and the organ in which it is assimilated (roots, root nodules, and leaves), the key enzymes involved in the assimilation are glutamine synthetase (GS) and glutamate synthase (GOGAT; glutamine-oxoglutarate aminotransferase), which are present in the roots, shoots, and N_2_-fixing organ [35,48,49]. Wang et al. [50] found 42% lower GS activity in wheat fed with NH_4_^+^ in acidic conditions (pH = 5.0) compared with those in more alkaline conditions (pH = 6.5). In coffee plants, Carr et al. [47] reported that the nitrogen form significantly influences the amino acid profile, with some of these amino acids being directly involved in stress signaling, like cysteine, which is very reactive and, therefore, toxic if it accumulates at a high proportion (>35% of the total intracellular content), with a gradual increase in cysteine concentration according to the NH_4_^+^ levels in the nutrient solution. Similar results were reported by the authors with regard to arginine, the levels of which were 1122% higher in plants subjected to the treatment with 100% NH_4_-N than with 100% NO_3_-N.

Regarding the influence of the interaction of N forms and water level on Ps, in coffee plants growing in a nutrient solution, the decrease in the QA reduction for plants grown in 0% NO_3_-N/100% NH_4_-N and 100% NO_3_-N/0% NH_4_-N was related to “closed” PSII centers, reflecting an accumulation of reduced QA and also non-photochemical energy dissipation. At the same time, these treatments showed lower photosynthesis rates, indicating higher non-photochemical dissipation as a result of the non-utilized energy, while the other treatments, with NO_3_/NH_4_ ratios of 50% NH_4_-N/50% NO_3_-N and 12.5%/NH_4_-N/87.5% NO_3_-N, exhibited greater efficiency in employing the absorbed light in the photochemistry process [47].

Under water-stress conditions, N application in *Coffea canephora* Pierre brought about an increase in cell-wall rigidity and osmotic adjustment, improving water extraction from drying soil in addition to avoiding the excessive loss of cell volume, thus leading to some degree of drought tolerance [20]. Moreover, N increases the long-term WUE through changes in photosynthesis [20]. Nitrates play a key role in root hydraulic resistance; for instance, root hydraulic resistance increases when NO_3_^−^ availability is low and decreases when NO_3_^−^ supply is high [18,51]. NO_3_^−^ uptake and assimilation involve a net consumption of protons, raising the possibility of direct feedback between NO_3_^−^ assimilation and the regulation of aquaporins [18,19,51]. Despite the existence of biochemical and biophysical mechanisms for pH homeostasis, blocking NR can lead to a measurable change in cytosolic pH, and a decrease in cytosolic pH has also been shown to reversibly alter root hydraulic properties due to the protonation of a tyrosine residue on the cytosolic side of the majority of plasma membrane intrinsic proteins (PIP), resulting in a dramatic increase of root hydraulic resistance [51,52]. This gating mechanism raises the possibility that changes in cytosolic pH due to NO_3_^−^ assimilation could be involved in triggering nitrate-induced changes in the permeability of the roots to water [51]. Meanwhile, a high NH_4_^+^ (3 mM) supply induced more apoplastic barrier formation and decreased root hydraulic conductivity when compared with low NH_4_^+^ supply (0.03 mM) in rice seedlings [19].

Proline (Pro) has been recognized as an essential amino acid derived from nitrogen synthesis, with glutamate as one of the main precursors. Pro accumulation is believed to play adaptative roles in plant-stress tolerance; it has been reported as a compatible osmolyte and a way to store carbon and nitrogen. Pro can be an ROS scavenger and has been proposed to function as a molecular chaperone, stabilizing the structure of proteins, and its accumulation can provide a way to buffer cytosolic pH and to balance cell redox status. Finally, Pro accumulation may be part of the stress signal influencing adaptative responses [53]. In coffee, Pro accumulation during stress conditions has been highly documented as a stress-tolerance mechanism [54,55]. The total content of amino acids, including Pro, increases gradually with an increase in nitrogen rates, while a higher NO_3_^−^ concentration in soils suppresses the synthesis of amino acids, including Pro, in plants exclusively fed with NO_3_^−^ fed compared with those with fed with a balance of NH_4_^+^ and NO_3_^−^ [56].

In well-created agricultural soils, mineral N and especially NO_3_^−^ are the most abundant forms of available N, while NH_4_^+^ dominates in soils which nitrification is inhibited (Von Wiren et al. [37]). In a short greenhouse trial (4 months) with coffee seedlings, using a high-organic-matter soil (160 g kg^−1^ of organic matter) as the growth medium, and using ^15^N-labeled urea, Salamanca et al. [57] did not find any impact of the interaction between soil N and moisture factors on the N uptake and recovery, likely because the moisture levels evaluated were 122, 100, 80, and 61% of the FC, indicating relatively high soil moisture levels that potentially inhibit the nitrification process. This is completely different to the findings of our study, where the effect of the interaction between the water level and N forms was significant for the N uptake in both growth media, namely quartz sand and soil (Table 2).

### 3.2. Synergy between Ammonium and Nitrate Nutrition

Ammonium is preferentially taken up by many plants when supplied in an equimolar concentration with nitrates, particularly when the N supply is low, and only in some low-temperature conditions can the NH_4_^+^ uptake continue while the NO_3_^−^ uptake is suppressed. Ammonium is assimilated through the roots, imposing a direct demand for carbon skeletons, which is reflected in higher activity levels of PEP carboxylase [35,39]. The increased carbon consumption caused by increased NH_4_^+^ assimilation in the roots also partially explains NH_4_^+^-induced growth inhibition [42]. Compared with NH_4_^+^, NO_3_^−^ has the advantage of allowing the more flexible distribution of assimilation between roots and shoots and can be stored in higher amounts than NH_4_^+^ in the vacuoles [35]. On the other hand, NH_4_^+^ and NO_3_^−^ comprise about 80% of the total cations and anions taken up by plants, and the form of N has a strong impact on the uptake of other cations and anions, as well as on cellular pH regulation and on rhizosphere pH [35].

The assimilation of NH_4_^+^ through the roots produces about one proton per molecule of NH_4_^+^, and the generated protons are to a large extent excreted into the external medium to maintain cellular pH and electro-neutrality, with the latter compensating for the excess uptake of cation equivalents over anion equivalents, which are generally associated with NH_4_^+^ nutrition [35,42]. Wang et al. [50] found that the growth of wheat seedlings was seriously inhibited by NH_4_^+^ feeding, and that this inhibitory effect was enhanced by soil-acidification stress. Excessive NH_4_^+^ accumulation has an energy cost related to providing energy for H+ pumping, which is expressed by a higher ATPase activity [42,58,59], as was observed in coffee plants treated with NH_4_^+^, which showed significantly higher ATP when subjected to treatment with 100% NH_4_/0% NO_3_ compared to the others with less ammonium and more nitrates [47]. Under mixed N nutrition, the protons generated by NH_4_^+^ assimilation can be used for NO_3_^−^ reduction; therefore, it is easier for plants to regulate their intracellular pH when both forms of nitrogen are supplied [35].

As reported by Vaast et al. [38], the concurrent uptake of NH_4_^+^ and NO_3_^−^ in coffee plants helps in the maintenance of the cation–anion balance within the root cells, thus minimizing the plant’s need for organic acid synthesis to regulate its intracellular pH. Direct NH_4_^+^ uptake from the solution decreases the energy cost involved in NO_3_^−^ reduction and increases the supply of reduced N for protein synthesis [60,61]. The current absorption of NH_4_^+^ and NO_3_^−^ also prevents acidification or alkalinization in the rhizosphere, which can in return affect the uptake [62].

In coffee, the low DW accumulation was not related to the low pH and was more related to the inhibition generated by the higher NH_4_-N or NO_3_-N content in the soil, because in both trials using quartz sand and soil, the soil pH increased linearly with the share of NO_3_-N in the nutrient solution, but the DW and Ps in Trials 1 and 3 were higher in the treatments with 75% NH_4_-N/25% NO_3_-N and 50% NH_4_-N/50% NO_3_-N, while in quartz sand, better physiological performance was achieved in the treatments with 100% NO_3_-N, regardless of the water level. This means that the concentrations of NO_3_^−^ and NH_4_^+^ in the soil are more important, and soil pH indirectly influences the nitrification rates, with a direct influence on the NH_4_^+^–NO_3_^−^ balance, which has a significant effect on nutrient uptake, and nitrogen assimilation.

### 3.3. Effect of the Interaction between N Forms and Water Level on Nutrient Uptake

In all three greenhouse trials, the highest nitrogen uptakes were observed in the treatments that had a balance of nitrogen forms (50% NH_4_-N/50% NO_3_-N) without water stress; under water stress, high N uptake was achieved by the treatments with more nitrates (25% NH_4_-N/75% NO_3_-N) in quartz sand and in the treatment with more NH_4_^+^ (100% NH_4_-N/0% NO_3_-N) in organic soil. Vaast et al. [38] reported similar NH_4_^+^ and NO_3_^−^ uptake rates in a solution culture with a pH range of 4.25 to 5.75 and noted that the total N uptake at any NH_4_/NO_3_ ratio was higher than that of plants fed solely with either NH_4_^+^ or NO_3_^−^.

As was reported by Carr et al. [47], the differences in growth and nutrient absorption displayed by the coffee plants in the greenhouse trials were directly correlated with the influence of the N forms, and varying the NH_4_-N/NO_3_^−^N ratios directly affected the charge balance of young coffee plants. A higher content of NH_4_-N in the nutrient solution increases the NH_4_^+^ content in the soil, with a stronger effect under water-stress conditions directly affecting the nutrient uptake. Excessive NH_4_^+^ feeding results in excessive acidification of the growth medium or nutrient solution, nutrient imbalance, and impaired plant growth.

The growth medium (quartz sand and soil) and the water level (with and without water stress) affect the contents of different N forms in the medium (Table 3), directly affecting the DW accumulation, Ps, and chlorophyll content in the leaves (Table 1 and Table 2; and Figure 3 and Figure 4). Under water-stress conditions, the NH_4_^+^ and NO_3_^−^ content in the soil was higher compared with that with the same treatments without water stress (Table 3). Regardless of the soil moisture content, the highest N and cation uptake was achieved when the concentration of NO_3_^−^ in the soil ranged between 1.5 and 3.5 mg 100 g^−1^ and that of NH_4_ was <0.3 mg 100 g^−1^.

According to Pilbeam and Kirkby [39], a variety of factors are involved in differences in the uptake and distribution of inorganic anions and cations in plants supplied with two forms of nitrogen. One of the most important of these factors is the alkalinization or acidification of the rhizosphere, as under acid conditions, phosphate becomes less available in the soil, the concentration of aluminum and manganese increases in the soil, and the uptake of calcium, magnesium, and potassium is depressed by H^+^ ions in the rhizosphere. 

For plants grown at the same pH but treated with either NO_3_-N or NH_4_-N, the uptake of calcium, magnesium, and potassium is higher for the NO_3_-fed plants [47], as was observed in this research (Table 3). In the case of plants with a high calcium demand like coffee, several disorders are generated by the shortage of calcium, namely reducing calcium translocation and mobility to new growth tissues, generating a reduction in calcium accumulation in mesophyll cells, with several effects on leaf physiology, and, together with a shortage of potassium and magnesium, significantly reducing plant growth and productivity [15].

The detrimental effects of NH_4_^+^ in reducing the uptake of potassium and calcium are likely to be more common when the pH of the growth medium is very low [36,43]. This indirect suppression of calcium and magnesium cation uptake indirectly reduces the nitrogen assimilation via low NR activity [63,64].

### 3.4. Influence of the N Forms on Productivity and Chlorophyll Content at Field Level

The differences in productivity effects between ammonium nitrate (AN) base fertilizers and urea are not always consistent, as different authors have reported that there does not exist significant differences on productivity between urea and AN base fertilizer in coffee [23,30,65], as we found during the first 2 years of harvest and after 3 years of treatment (Figure 5).

In perennial crops, it is common not to find a response to mineral N fertilization during the first few years; for example, in citrus in Brazil, Cantarella et al. [66] compared two N forms and did not find any response to mineral N fertilization during the first 3 years when comparing urea and AN. This lack of response was attributed by the authors to the high N reserves associated with the previous years of crop growth, but, after the 4th year of the trial, the citrus plantation responded to the mineral N fertilization, and differences between N sources were observed, with a significant higher yield for the treatments with AN compared with urea at the same N rate.

The low yield provided by the urea treatments is also related to the low efficiency of this N form. Of the total amount of mineral N fertilizer applied, only 25–60% of the mineral N is effectively taken up by coffee plants [67,68,69]. In coffee, for example, 20% of the mineral nitrogen applied was found in the shoots or aerial parts, 12% was found in the roots, and 26% was exported during harvest, while about 15% remained in the soil [68]. The rest is lost from the soil–plant system through mechanisms such as denitrification, leaching from the root zone, and volatilization. [5,22,30,70,71].

Without any soil or environmental limitations, the ammonification and nitrification of the nitrogen fertilizers take place in short periods of time [72], meaning that the nitrogen lixiviation of urea or AN base fertilizer in coffee is the same under field conditions [30,31].

In coffee-production systems in Colombia, Leal et al. [29] found a mean nitrogen volatilization (NH_3_-N) of 30% to 35% of the total N applied using urea. In a long-term trial in coffee during four crop seasons in Brazil, de Souza et al. [23] found that the NH_3_-N volatilization was significantly influenced by the nitrogen rate and form, where urea led to the highest values of NH_3_-N, ranging between 9% and 25%, varying with the rate and application time, while the losses associated with ammonium nitrate represented less than 1% of the doses applied in the four crop seasons.

The incorporation of fertilizer into the soil solution and exchange system is directly influenced by the rainfall volume and intensity; higher rainfall events are usually associated with better fertilizer incorporation and, consequently, lower losses by volatilization. A lower precipitation volume immediately after the application of the fertilizer to the soil hinders the incorporation of N and favors NH_3_-N losses [30]. Moreover, the architecture of coffee plants obstructs the direct incidence of rainfall that would incorporate the applied N fertilizer into the soil under the canopy [73,74,75]. Additionally, the presence of plant residues in the soil also acts as a barrier to fertilizer incorporation and creates a favorable environment for volatilization due to the high concentration and activity of urease [30,76].

Thus, the use of ammonium nitrate (AN) as a N source contributes the most to increasing soil N stocks and supplying the demands of the vegetative and reproductive stages of coffee plants, without any interference in the soil’s microbial and enzymatic activity [30]. The lower NH_3_-N losses from the use of AN can increase the average mineral N content by 50% when compared to urea and urea with inhibitors [77].

It is common to find in short-term field studies, not only in coffee, that the yields may not be significantly affected, despite the NH_3_-N losses [24,31], as was also found in this study during 3 years of treatment application and two of harvest (Figure 5). This confuses agronomists and farmers, but the logic of these results is well explained by Cantarella, [31]: “Soil is the major supplier of nutrients for plants, including N. A substantial part of the N that plants absorb during the production cycle come from the soil; fertilizers supplement the needs of crops. Thus, losses are canceled by soil supply and are not always reflected in yields in the short term. The N losses by volatilization is dispersed in the atmosphere and does not recharge the soil stock, the one that provides the nutrient for the plant. Over time, the soil becomes impoverished and N runs out. Under these conditions, the differences between sources subject to or not subject to volatilization become apparent”.

When the N and C stocks in the soil are low or do not exist, differences between N forms are rapidly observed, as was demonstrated by Chagas et al. [28] in coffee seedlings grown in an acidic Red Latosol soil collected in the B horizon with a very low level of organic matter (0.16%), who found significantly higher agronomic efficiency in plants fed with AN compared with urea and urea with inhibitors like formaldehyde or polyurethane.

In the present research, the differences in productivity effects between N forms could not be directly associated with soil N or C depletion due to the fact that the foliar N content after 4 years of treatment application was not significantly different among the N treatments (Table 4). The significant effect of a higher NO_3_^−^ share on productivity could be more closely related to the better nitrogen and cation uptake compared with NH_4_^+^-rich treatments including urea. Urea, as an organic N form, must be catalyzed into NH_4_^+^ before it can be absorbed by plants [78]. In other words, NH_4_^+^ and urea share a common metabolic pathway [43]. In our research, after 5-to-8 months in the greenhouse and 4 years in the field, the results indicate that coffee growth, Ps, nitrogen assimilation, nutrient uptake (N and cations), chlorophyll content, and productivity are affected by NH_4_^+^ and urea application, and the main effects of long-term NH_4_^+^ and urea feeding could be linked to the moderated inhibition of NO_3_^−^ uptake by NH_4_^+^ [38] and the disruption of the ion balance that strongly limits the N and cation uptake, with a subsequent influence on Ps, chlorophyll accumulation, and finally on productivity.

## 4. Materials and Methods

Over a 5-year period (2017 to 2022), three experimental trials under greenhouse conditions and one under field conditions were established with the aim of evaluating the effect of different forms of nitrogen and their interaction with water stress on the physiological and agronomic responses of coffee plants.

### 4.1. Greenhouse Trials

The greenhouse trials were carried out in Dülmen, Germany at the Hanninghof Research Center of Yara International. The mean air temperature was 23.1 °C (±2.2 °C), with a maximum air temperature of 31.4 °C and a minimum air temperature of 15.3 °C, a mean relative humidity of 64% (±10%), and a mean light intensity of 20.0 Klux during summertime. Supplemental light (300 mmol m^−2^ s^−1^ photosynthetic photon flux density) over 12–14 h was provided when the natural light became insufficient.

Arabica coffee (*Coffea arabica* L.) seeds of the varieties Topazio MG 1190 (Trial 1) and Castillo^®^ (Trials 2 and 3) were pre-germinated in dark conditions, and before the radicle emerged (BCH scale 03; [79]), the pre-germinated seeds were moved to small containers with perlite as the growth medium. The seeds were allowed to germinate until the plants reached three pairs of leaves that were completely open (BBCH Scale 13). During the germination process, the plants received a nutrient solution once per week containing N (7.6 mM), P (0.3 mM), K (1.7 mM), Mg (0.2 mM), Ca (0.9 mM), Fe (5.0 mM), Mn (2.9 mM), Zn (1.5 mM), Cu (0.6 mM), B (9.2 mM), and Mo (0.2 mM). Once the plants reached BBCH Scale 13, they were transplanted into 4.5 L pots containing quartz sand and/or soil as the growth medium according to the trial setup; during this growing phase, the plants received a nutrient solution once per week containing P (1.3 mM), K (14.3 mM), Mg (3.5 mM), S (1.2 mM), Ca (5.5 mM), Fe (14.7 mM), Mn (8.4 mM), Zn (5.3 mM), Cu (14.2 mM), B (5.2 mM), and Mo (1.5 mM).

The aim of the greenhouse trials was to test the physiological responses of young coffee plants to different nitrogen forms and their interaction with two water levels (with and without drought stress). Five N forms were analyzed as follows: 100% NH_4_-N/0% NO_3_-N, 75% NH_4_-N/25% NO_3_-N, 50% NH_4_-N/50% NO_3_-N, 25% NH_4_-N/75% NO_3_-N, and 0% NH_4_-N/100% NO_3_-N. Nitrogen forms were applied to the nutrient solution once per week using a N concentration between 21.4 and 28.6 mM according to the plant-development stage, combining sources containing NH_4_-N and NO_3_-N. A total of three greenhouse trials were conducted. Trial 1 evaluated the influence of the N forms on growth, nutrient uptake and soil acidity changes using soil as the growth medium. Trials 2 and 3 evaluated the effect of the interaction of N forms and water levels on growth, nutrient uptake, soil acidity, photosynthesis, and chlorophyll content using quartz sand and soil as the growth media, respectively.

Two water levels were set up according to the water holding capacity of the growth medium, known as the field capacity (FC). Treatments without water stress kept the soil moisture between 55 and 60% of the FC, and treatments with water stress kept the soil moisture between 35 and 40% of the FC. In Trial 1, the plants were transplanted in May 2018 and harvested in April 2019 after growing for 10 months. In Trial 2, the plants were transplanted in July 2020, and 2 months after transplantation, the reduction in the application of water in the water-stressed treatments began, with a duration of 3 months, and finally, the plants were harvested 5 months after transplantation in December 2020. In Trial 3, the plants were transplanted in July 2021 and 2 months after transplantation, the reduction in the application of water in the water-stressed treatments began, with a duration of 6 months; finally, the plants were harvested 8 months after transplantation.

To determine the FC of the pots, the substrates (soil and quartz sand) were saturated with water and pots were covered with black plastic to avoid evaporation. Once the free drainage stopped, the weight of the pots was measured and considered as the point of FC. For all three trials, the gravimetric soil moisture content was measured daily and was adjusted according to the water level: for Treatment 1, the soil moisture was adjusted to 60% of the FC daily, and for Treatments 2 and 3 with and without water stress, it was adjusted as was described previously. For Trials 1 and 3, a loamy sandy soil was used as the growth medium, with the following characteristics: pH, 5.76; organic carbon, 2.1%; P, K, Ca, and Mg contents, 56.3, 15.16, 57.2, and 6.98 mg.100 g^−1^, respectively. The pH was determined using 0.01 M CaCl_2_; organic carbon was determined via the dry combustion method; P content was determined using Mehlich 3 (0.2 M CH_3_COOH, 0.25 M NH_4_NO_3_, 0.015 M NH_4_F, 0.013 M HNO_3_, and 0.001 M EDTA); and K, Ca, and Mg contents were determined by means of 1 N ammonium acetate extraction (1 N NH_4_C_2_H_3_O_2_, pH 7.0). N forms (NH_4_-N and NO_3_-N) in the growth medium were estimated using the continuous flow analysis (CFA) method.

The photosynthesis rates (Ps) in Trials 2 and 3 were measured 2 months after water-stress application, selecting sunny days with a mean air temperature of 23 to 25 °C, and readings were performed between 10:30 a.m. and 12:30 p.m. randomly among treatments, selecting four pairs of fully expanded leaves from the apex to the stem in the middle part of the plant using the li-6400 Portable Photosynthesis System (Li-Cor Bioscience, Inc., Lincoln, NE, USA). The equipment was set up with the following parameters: temperature = 25 °C; CO_2_ concentrations = 370 μmol mol^−1^, PAR = 1000 μmol photons m^−2^ s^−1^. Chlorophyll meters are widely used for diagnostics and monitoring the nitrogen nutrition status of foliage in many crops through rapid, non-destructive measurements [80,81]. In this research, we used the N-Tester (Yara International, Oslo, Norway) to measure the chlorophyll index.

At harvest time in each of the trials, the coffee plants were trimmed and the tissues (roots, leaves, stems, and branches) were dried in an oven at 65 °C until a constant weight was attained. The dried material was then finally ground for nutrient analysis in the lab. After tissue harvest, growth medium samples (soil and quartz sand) were taken for each replicate and oven dried at 100 °C until a constant weight was attained and then were made uniform using 2 mm meshes.

Finally, milled plant materials were used for elemental analysis after wet digestion in a microwave digester (MLS mega; MLS GmbH, Leutkirch, Germany). All of the micro- and macronutrients (excluding nitrogen) were analyzed using inductively coupled plasma optical emission spectrometry (Perkin–Elmer Optima 3000 ICP-OES; Perking-Elmer Corp, Shelton, CT, USA). The nitrogen was determined by means of the micro-Kjeldahl method.

The experiments were set up in a completely randomized manner for Trial 1, while a factorial experimental design was used for Trials 2 and 3, with 10 replicates per treatment.

### 4.2. Field Trial

The field trial was conducted in the central west region in the municipality of Manizales in the department of Caldas–Colombia during the period from January 2017 to December 2020. The study area possesses a soil classified as an Inceptisols–Typic Eutropepts (USA soil taxonomy), with metamorphic rock as a parental material, with a Superficial A horizon covered by layers of volcanic ashes of different thickness, and a clayey reddish-yellow B horizon [82,83].

The trial was performed at a commercial coffee farm “Naranjos” located at 5°01′ N–75°20′ W at 1400 m elevation. The soil conditions at a depth between 0 and 20 cm were as follows: pH, 4.76; organic carbon, 3.85%; and K, Ca, Mg and Al content, 1.08, 3.25, 1.17, and 1.14 Cmolc.Kg^−1^, respectively, with a soil particle distribution of 40% sand, 40% loam, and 20% clay. The pH was determined in water 1:1; organic carbon was determined by means of the Walkley–Black method; the exchangeable fraction of K, Ca, and Mg was determined with 1 N ammonium acetate; the Al content was determined by means of the KCl extraction method; and soil texture analysis was performed using the hygrometer Bouyoucos method, with the mean water balance parameters described in Table 5.

The coffee variety used was *Coffea arabica* L Variety Castillo^®^, established in 2007 in full sunshine conditions (FS), planted at a density of 6500 plants ha^−1^ with 1.1 m between plants and 1.4 m between rows. Before treatment application, the plantations were stem trimmed at a height of 30 cm in January 2017, aiming to rejuvenate the plantation and to initiate a new productive cycle from 2018 to 2020.

A total of five treatments were evaluated per trial, consisting of four nitrogen forms, namely urea, 75% NH_4_-N/25% NO_3_-N, 50% NH_4_-N/50% NO_3_-N, and 25% NH_4_-N/75% NO_3_-N, along with one control group without nitrogen, placed in a completely randomized block experimental design with four replicates. Each plot in the block had an area of 28 m^2^ with 18 effective plants. Four plants in each plot were used for the assessment at harvest during each year and the chlorophyll index measurements at intervals of 2 months using the N-tester. The yield assessment was performed by picking each plot by hand over the course of the whole year. For the region where the trial was located, 75% of the yearly harvest was collected between September and December, and 25% was collected between March and June.

The N rates were the same for all of the treatments using the growing rates for each year according to the crop demand per season (Table 6). The nitrogen sources used were urea (46% N-100% Ureic); calcium nitrate (16% N-8% NH_4_-N/92% NO_3_-N); ammonium sulphate (21% N-100% NH_4_-N/0% NO_3_-N); ammonium nitrate base NPK (27% N-61% NH_4_-N/39% NO_3_-N); ammonium diphosphate (18% N-100% NH_4_-N/0% NO_3_-N); and potassium nitrate base NPK (14% N-75% NH_4_-N/96% NO_3_-N). Other nutrients were compensated for using single products without N, such as magnesium sulfate or kieserite, (MgSO4), magnesium oxidate (MgO), potassium muriate (MOP), gypsum, boric acid, and triple super phosphate (TSP).

Daily rainfall conditions were measured during the study period; the soil moisture index (SMI) was calculated following the methodology proposed by Ramirez et al. [84]. Water deficit was calculated as follows: if SMI < 0.4, water deficit = crop evapotranspiration (ET_C_); if SMI > 0.4, water deficit = 0. The water excess was calculated as follows: if SMI > 0.69, water excess = actual soil moisture − soil moisture at field capacity; if SMI < 0.69, water excess = 0.

Phytosanitary control was applied only for coffee berry borers (*Hyphotenemus Hampey*, Ferrary, Curculionidae), using the farmer practices for all the field, based on low-toxicity insecticides and biological control with entomopathogenic fungus.

Statistical analysis was performed according to the trial setup, and the analysis of variance (ANOVA) was estimated according to the experimental design using the Statgraphics Centurion XV software package (Statgraphics Technologies, Inc., The Plains, VA, USA). The Shapiro–Wilks modified test was applied for the normality test and the residual-versus-prediction test to evaluate the heterogeneity of variances. Fischer’s test was used to detect the treatments significantly affecting the ANOVA. 

## 5. Conclusions

In coffee production, nitrogen forms significantly influence the physiological and agronomical responses of coffee plants and can be considered as important alternatives for improving the adaptation capacity of this crop to drought stress. After three greenhouse trials and one field trial, we can draw the following conclusions: Coffee is susceptible to drought stress; when the plants grow in limited water conditions for a period of 3 months, there is a significant reduction of the photosynthesis capacity (Ps), an increase in the chlorophyll content, and a reduction in the biomass accumulation.Nitrogen forms interact significantly with the water levels; under water-stress conditions, plants fed with a balance of NH_4_-N/NO_3_-N show significantly higher photosynthesis rates and chlorophyll contents compared with those plants fed with 100% NH_4_^+^ or 100% NO_3_^−^.A balance between nitrogen forms in coffee production improves the nitrogen and cation uptake; in all the greenhouse trials, treatments with 100% NH_4_^+^ significantly reduced the N and cation uptake, while higher N, Ca, and Mg uptake was achieved in the treatments with a balance of 50% NH_4_-N/50% NO_3_-N and 25% NH_4_-N/75% NO_3_-N.In organic soils, plants fed with 100% NO_3_^−^ significantly display reduced nitrogen uptake and incorporation with a significant reduction in chlorophyll and biomass accumulation, suggesting that, before applying nitrogen in coffee, the soil content of NO_3_^−^ should not exceed 10 mg 100 g^−1^.At the field level, a balance of 25% NH_4_/75% NO_3_ improves nitrogen assimilation through higher chlorophyll formation, with a subsequent yield improvement with respect to NH_4_^+^-rich N forms, including urea.At the field level, the chlorophyll content was positively correlated with the N content in the leaves and with the productivity.There is a need for further research to evaluate the influence of N forms on N lixiviation and nitrous oxide emissions at the field level under variable soil and climate conditions.

## Figures and Tables

**Figure 1 plants-13-01387-f001:**
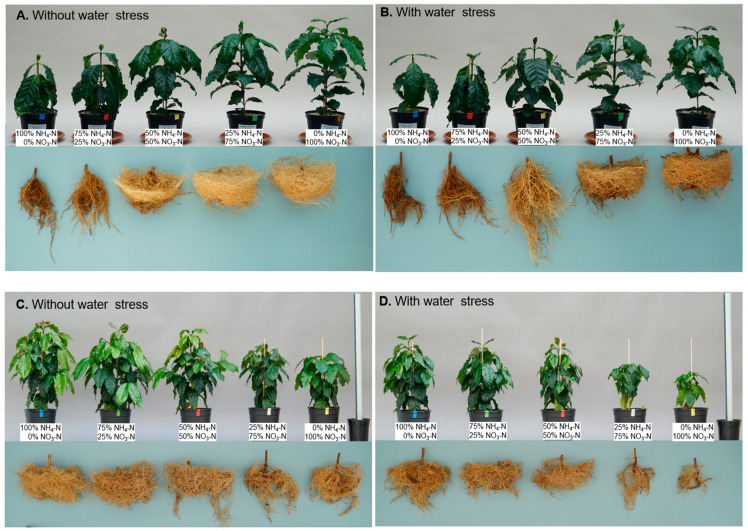
Influence of the water level and nitrogen forms on coffee plant growth and development. Quartz sand without water stress (**A**), quartz sand with water stress (**B**), soil without water stress (**C**), and soil with water stress (**D**).

**Figure 2 plants-13-01387-f002:**
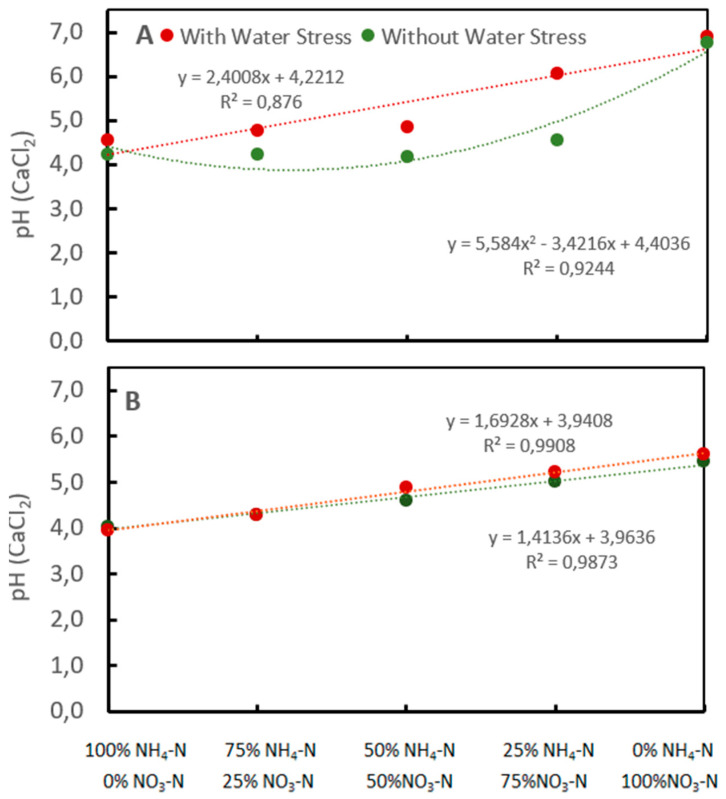
Influence of the water level and N forms on soil acidity. Quartz sand (**A**). Soil (**B**).

**Figure 3 plants-13-01387-f003:**
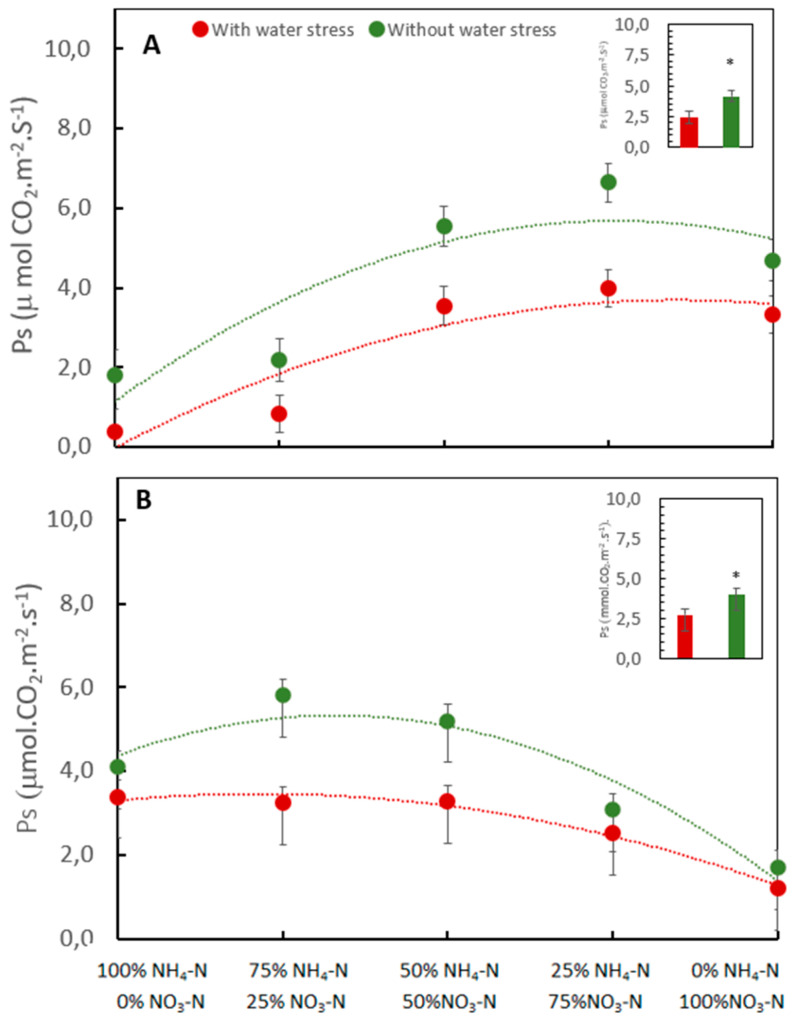
Influence of the N forms and water-stress conditions on photosynthesis (Ps) in coffee plants growing in two substrates. Quartz sand (**A**). and soil (**B**). * *p* value < 0.05 according to Fisher´s LSD test.

**Figure 4 plants-13-01387-f004:**
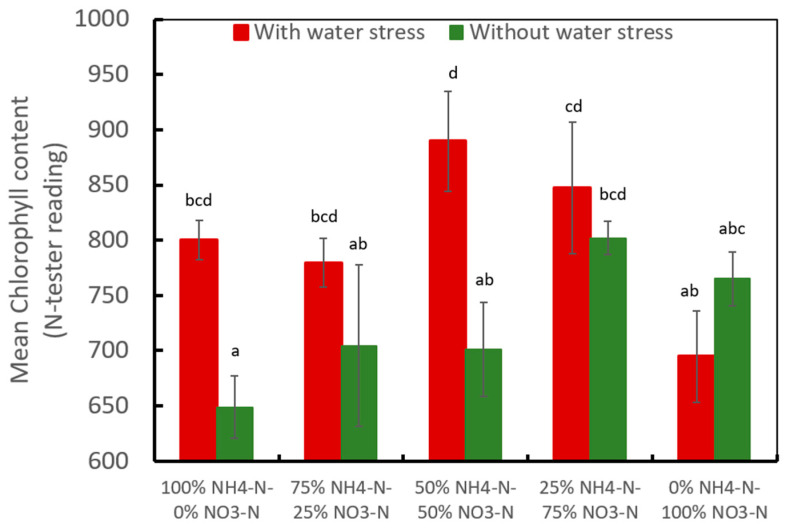
Influence of the water level and N forms on the chlorophyll content in mature coffee leaves in coffee plants growing in soil (greenhouse trial 3). Different letters denote statistically significant differences according to Fisher’s LSD test alpha = 0.05.

**Figure 5 plants-13-01387-f005:**
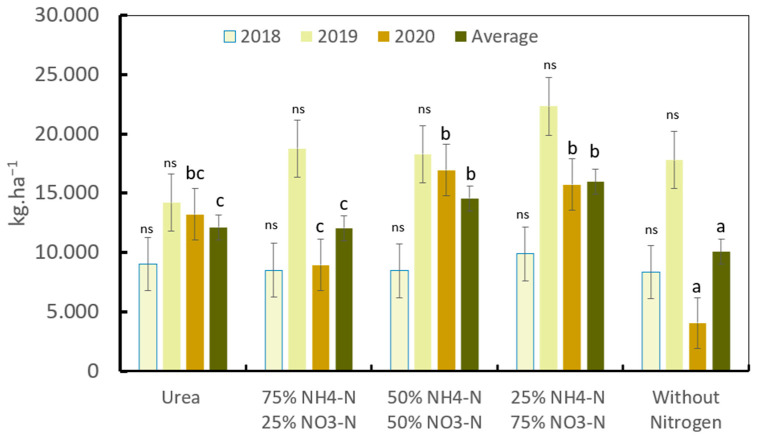
Influence of different nitrogen forms on coffee productivity after stem trimming. Different letters denote statistically significant differences according to Fisher’s LSD test alpha = 0.05; ns: not significantly differences.

**Table 1 plants-13-01387-t001:** Effect of N forms and water levels on dry biomass accumulation in coffee.

Trial	N Form	Without Water Stress			With Water Stress		
		Shoot DW	Root DW	Total DW	Shoot DW	Root DW	Total DW
		g Plant^−1^					
1	N forms in soil without water stress						
	100% NH_4_-N/0% NO_3_-N	21.7 ^ab^	4.5	26.2 ^a^	--	--	--
	75% NH_4_-N/25% NO_3_-N	34.4 ^c^	6.9	41.3 ^b^	--	--	--
	50% NH_4_-N/50% NO_3_-N	32.8 ^bc^	6.4	39.2 ^ab^	--	--	--
	25% NH_4_-N/75% NO_3_-N	26.6 ^abc^	4.5	30.1 ^ab^	--	--	--
	0% NH_4_-N/100% NO_3_-N	19.6 ^a^	4.2	23.8 ^a^	--	--	--
	N Form	**	ns	**			
2	N forms in quartz sand without and with water stress						
	100% NH_4_-N/0% NO_3_-N	6.7 ^ab^	1.2 ^a^	7.9 ^a^	6.4 ^a^	1.7 ^b^	8.1 ^a^
	75% NH_4_-N/25% NO_3_-N	9.4 ^c^	3.9 ^de^	13.2 ^c^	7.3 ^b^	4.4 ^ef^	11.5 ^b^
	50% NH_4_-N/50% NO_3_-N	14.6 ^f^	4.9 ^f^	19.6 ^e^	11.2 ^d^	1.2 ^a^	12.5 ^bc^
	25% NH_4_-N/75% NO_3_-N	17.5 ^g^	1.5 ^ab^	19.5 ^e^	12.8 ^e^	2.7 ^c^	16.4 ^d^
	0% NH_4_-N/100% NO_3_-N	17.8 ^g^	3.6 ^d^	21.9 ^f^	13.1 ^e^	3.5 ^d^	16.9 ^d^
	N Form	***	***	***			
	Water Level	***	**	***			
	N Form × Water Level	***	***	***			
	N forms in soil without and with water stress						
3	100% NH_4_-N/0% NO_3_-N	30.8 ^cd^	33.8 ^c^	44.3 ^c^	22.7 ^bc^	9.5 ^ab^	26.5 ^b^
	75% NH_4_-N/25% NO_3_-N	35.4 ^d^	29.3 ^c^	47.1 ^c^	21.3 ^b^	9.5 ^ab^	25.0 ^b^
	50% NH_4_-N/50% NO_3_-N	31.2 ^d^	25.8 ^bc^	41.5 ^c^	15.3 ^ab^	8.0 ^a^	18.5 ^ab^
	25% NH_4_-N/75% NO_3_-N	20.7 ^b^	7.0 ^a^	23.4 ^b^	11.8 ^a^	4.2 ^a^	13.4 ^ab^
	0% NH_4_-N/100% NO_3_-N	14.5 ^ab^	4.8 ^a^	16.4 ^ab^	7.6 ^a^	1.8 ^a^	8.3 ^a^
	N Form	***	**	***			
	Water Level	***	**	***			
	N Form × Water Level	ns	ns	ns			

Different letters denote statistically significant differences according to Fisher’s LSD test alpha = 0.05; ** *p* value < 0.01; *** *p* value < 0.001; ns: not significantly differences.

**Table 2 plants-13-01387-t002:** Effect of the interaction of N forms with water levels on nutrient uptake.

Trial	N Form	Without Water Stress				With Water Stress			
		N	K	Ca	Mg	N	K	Ca	Mg
		g Plant^−1^							
N forms in soil without water stress								
1	100% NH_4_-N/0% NO_3_-N	0.887 ^b^	0.588 ^a^	0.078 ^a^	0.047 ^a^	--	--	--	
	75% NH_4_-N/25% NO_3_-N	1.236 ^c^	0.926 ^c^	0.123 ^a^	0.076 ^bc^	--	--	--	
	50% NH_4_-N/50% NO_3_-N	1.165 ^c^	1.055 ^dc^	0.190 ^b^	0.086 ^cd^	--	--	--	
	25% NH_4_-N/75% NO_3_-N	0.965 ^bc^	1.089 ^d^	0.283 ^d^	0.081 ^d^	--	--	--	
	0% NH_4_-N/100% NO_3_-N	0.572 ^a^	0.768 ^b^	0.244 ^c^	0.056 ^a^	--	--	--	
	N Forms	***	***	***	***				
2	N forms in quartz sand without and with water stress								
	100% NH_4_-N/0% NO_3_-N	0.203 ^a^	0.101 ^a^	0.016 ^a^	0.011 ^a^	0.185 ^a^	0.099 ^a^	0.017 ^a^	0.010 ^a^
	75% NH_4_-N/25% NO_3_-N	0.435 ^c^	0.244 ^b^	0.028 ^b^	0.019 ^b^	0.335 ^b^	0.254 ^b^	0.031 ^b^	0.026 ^b^
	50% NH_4_-N/50% NO_3_-N	0.538 ^f^	0.433 ^d^	0.043 ^c^	0.039 ^c^	0.433 ^c^	0.313 ^c^	0.034 ^b^	0.037 ^c^
	25%MH4-N/75% NO_3_-N	0.464 ^cd^	0.463 ^de^	0.077 ^e^	0.052 ^d^	0.506 ^def^	0.484 ^e^	0.066 ^c^	0.057 ^d^
	0% NH_4_-N/100% NO_3_-N	0.530 ^ef^	0.558 ^f^	0.125 ^g^	0.072 ^e^	0.483 ^cde^	0.524 ^f^	0.109 ^f^	0.072 ^e^
	N Forms	***	***	***	*				
	Water Level	***	**	***	ns				
	N Form × Water Level	**	***	**	ns				
3	N forms in soil without and with water stress								
	100% NH_4_-N/0% NO_3_-N	0.683 ^cd^	0.579 ^d^	0.197 ^bc^	0.088 ^d^	0.807 ^de^	0.867 ^ef^	0.137 ^a^	0.071 ^c^
	75% NH_4_-N/25% NO_3_-N	0.826 ^e^	0.574 ^d^	0.295 ^d^	0.109 ^e^	0.735 ^cde^	0.894 ^f^	0.171 ^ab^	0.066 ^c^
	50% NH_4_-N/50% NO_3_-N	0.849 ^e^	0.385 ^bc^	0.329 ^d^	0.095 ^d^	0.516 ^b^	0.764 ^e^	0.179 ^ab^	0.051 ^b^
	25% NH_4_-N/75% NO_3_-N	0.675 ^c^	0.295 ^ab^	0.295 ^d^	0.066 ^c^	0.350 ^a^	0.548 ^d^	0.197 ^bc^	0.038 ^b^
	0% NH_4_-N/100% NO_3_-N	0.507 ^b^	0.196 ^a^	0.229 ^c^	0.044 ^b^	0.228 ^a^	0.412 ^c^	0.140 ^a^	0.023 ^a^
	N Forms	***	***	***	***				
	Water Level	***	***	***	***				
	N Form × Water Level	***	ns	ns	*				

Different letters denote statistically significant differences according to Fisher´s LSD test alpha = 0.05; * *p* value < 0.05; ** *p* value < 0.01; *** *p* value < 0.001; ns: not significantly differences.

**Table 3 plants-13-01387-t003:** Influence of water level and nitrogen forms on nitrogen content in the growth medium at harvest time in two growing substrates.

N Form	Quartz Sand ^¥^				Soil ^£^			
	Without Water Stress		With Water Stress		Without Water Stress		With Water Stress	
	NH_4_-N	NO_3_-N	NH_4_-N	NO_3_-N	NH_4_-N	NO_3_-N	NH_4_-N	NO_3_-N
	mg 100 g^−1^							
100% NH_4_-N/0% NO_3_-N	9.52 ^D^	0.02 ^a^	11.12 ^E^	0.04 ^a^	0.046 ^ab^	0.54 ^A^	0.274 ^de^	1.23 ^A^
75% NH_4_-N/25% NO_3_-N	3.91 ^B^	0.10 ^a^	5.51 ^C^	0.60 ^bc^	0.028 ^ab^	0.94 ^A^	0.352 ^e^	2.73 ^AB^
50% NH_4_-N/50% NO_3_-N	0.48 ^A^	0.02 ^a^	0.84 ^A^	0.29 ^a^	0.018 ^ab^	1.76 ^AB^	0.230 ^cd^	9.69 ^CD^
25% NH_4_-N/75% NO_3_-N	0.02 ^A^	0.03 ^a^	0.19 ^A^	0.89 ^c^	0.026 ^ab^	6.73 ^BC^	0.122 ^bc^	16.06 ^EF^
0% NH_4_-N/100% NO_3_-N	0.01 ^A^	0.30 ^ab^	0.03 ^A^	1.44 ^d^	0.008 ^a^	12.10 ^DE^	0.034 ^ab^	19.18 ^F^
N Forms	***	***			***	***		
Water Level	**	***			***	***		
N Form × Water Level	ns	***			*	ns		

Different letters denote statistically significant differences according to Fisher´s LSD test alpha = 0.05; * *p* value < 0.05; ** *p* value < 0.01; *** *p* value < 0.001; ns: not significantly differences. ^¥^ GH trials 2; ^£^ GH trials 3.

**Table 4 plants-13-01387-t004:** Influence of the N forms on the chlorophyll content in the 4th year after treatment applications.

N Form	February	April	June	August	October	Mean	Nitrogen
	N-Tester Reading						
Urea	947.0 ^defghi^	944.0 ^defgh^	887.0 ^cdef^	979.7 ^fghi^	973.5 ^fghi^	946.2 ^B^	2.68 ^b^
75% NH_4_-N/25% NO_3_-N	937.0 ^cdefg^	1005.7 ^ghi^	894.2 ^cdef^	1025.0 ^ghi^	1052.5 ^i^	982.9 ^BC^	2.69 ^b^
50% NH_4_-N/50% NO_3_-N	886.7 ^cdef^	1016.2 ^ghi^	989.5 ^fghi^	969.5 ^fghi^	964.0 ^efghi^	965.2 ^B^	2.70 ^b^
25% NH_4_-N/75% NO_3_-N	1015.0 ^ghi^	1043.5 ^hi^	1051.0 ^i^	1029.5 ^ghi^	976.2 ^fghi^	1023.0 ^C^	2.58 ^b^
Control (No-N)	725.0 ^a^	770.2 ^a^	856.5 ^bcd^	838.2 ^bc^	860.5 ^bcde^	810.1 ^A^	2.35 ^a^
Mean	902.1 ^A^	955.9 ^B^	935.6 ^AB^	968.4 ^B^	965.3 ^B^		
N Forms ***							
Month **							

Different letters denote statistically significant differences according to Fisher’s LSD test alpha = 0.05; ** *p* value < 0.01; *** *p* value < 0.001; ns: not significantly differences.

**Table 5 plants-13-01387-t005:** Rainfall distribution for the field trial *.

Year	Rainfall	Water Deficit	Water Excess
	mm		
2017	2321.0	0.0	976.3
2018	2262.0	149.6	249.9
2019	2232.0	178.4	341.5
2020 until June ^++^	902.0	48.4	87.1
Mean	1929.3	94.1	413.7

* Data from the rain gauge located in the farm. Water deficit and water excess calculated. ^++^ The rain gauge broke and the rainfall data could not be recorded for the remaining period.

**Table 6 plants-13-01387-t006:** Nutrient rates used in the field trials during the whole growing season.

Year	N	P_2_O_5_	K_2_O	CaO	MgO	S
	kg ha^−1^					
2017	124	38	140	83	23	62
2018	213	38	160	125	17	77
2019	316	91	314	114	45	92
2020	371	76	310	114	57	107
Mean	256	61	231	109	36	85

## Data Availability

Data are contained within the article.

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
