# Peer review of "Physiological and Agronomical Response of Coffee to Different Nitrogen Forms with and without Water Stress"

_plants, 2024, doi:10.3390/plants13101387_

Round 1

Reviewer 1 Report

Comments and Suggestions for Authors

Dear respected

Thanks too much of giving me the opportunity to evaluate the research paper: Physiological and Agronomical Response of Coffee to Different Nitrogen Forms with and without Water Stress.

 evaluate the influences of different NH4-N:NO3-N  rations in coffee from the physiological and agronomical perspective, and its interaction with soil  water levels. During five years, three trials were conducted under controlled conditions in greenhouse with different growing mediums (quartz-sand) and organic soil, with and without water stress and one trial under field conditions. However, The introduction needs to be rearranged and to show the aim of the research more clearly, and it also needs to be supported by some modern references. My mean concern is why the authors don't sufficient describe the determination of most analysis in the Material and method.. Moreover,

1- It is required to show the importance of the study in the introduction to the abstract and the most important recommendations must be highlighted at the end of the abstract, which justifies the practical importance of the study.
2- Scientific names must be in italic in the whole manuscript as well as in the references list.
3- The 'discussion' section is quite comprehensive; however, the authors should consider more about logic of the current work.
4- The tables and figures: Please improve the quality of tables and figures. The quality of all figures is not acceptable for an international journal.
5- I recommended to improve the conclusion section.
6- References need to be checked, because once you use the abbreviation of the name of the journal and once you write the full name of the journal, so please refer to the journal guidelines.

Comments on the Quality of English Language

Minor editing of English language required

Author Response

reply to Reviewer 1

1- It is required to show the importance of the study in the introduction to the abstract and the most important recommendations must be highlighted at the end of the abstract, which justifies the practi-cal importance of the study.
R/ New lines added 13 to 15 and 25-26.

2- Scientific names must be in italic in the whole manuscript as well as in the references list.
R/ Done

3- The 'discussion' section is quite comprehensive; however, the authors should consider more about logic of the current work. R/ adjusted

4- The tables and figures: Please improve the quality of tables and figures. The quality of all figures is not acceptable for an international journal.
R/ Done

5- I recommended to improve the conclusion section.
R/ Done

6- References need to be checked, because once you use the abbreviation of the name of the journal and once you write the full name of the journal, so please refer to the journal guidelines.
R/ Done

Reviewer 2 Report

Comments and Suggestions for Authors

This manuscript reported the changes in physiological and agronomical properties of coffee plants under different mineral N forms and water stress conditions using quartz-sand and soil media. The results show that a reasonable ratio of ammonium and nitrate can affect the physiological and agronomical properties of coffee plants, which will vary with water stress levels. Hence it has a value for publication. Some minor comments are shown at the bottom.

1) Line 10, changed into "N uptake by the roots is dominated by..."

2) Line 14, rations, changed into ratios

3) Line 20, changed into "has shown..."

4) Lines 22-25, this sentence is not clear. Please rephrase it.

5) Lines 36-37: what is the meaning of 90 million of 60 kg coffee bags?  

6) Line 44: The worldwide coffer production it is under threat..., changed into The worldwide coffer production is under threat.... In the whole text, some similar expressions are available.

7) Line 49, what is the meaning of -S?

8) Line 73: Mineral fertilization it is an important driver of the productivity ..., the same as 6). Please delete the word "it".

9) Lines 74-78 The sentence needs to be refined.

10) Lines 83-84, what is the meaning of the degree of fertilizer used?

11) Lines 100-101, the sentence is not clear.

12) Lines 118-126. this paragraph may be deleted. Anyway this introduction needs to be refined by removing unnecessary contents. 

13) In Table 1, the word P-value can be removed. At present significant differences between treatments?

14) In Figure 1, "...soil with water stress (B)", changed into "...soil with water stress (D)".

15) Line 227, ... using organic soil as a growth medium." However, in the other places, the authors used soil as a growth medium.

16) Table 3, in the soil, changed into growth medium. The descriptions for significant differences are not clear for capital and small letters. The same for Figures 4 and 5 and other tables or Figures. Also, significant differences in Table 4 are indeed not unknown. 

17) 2.4 section, nitrogen forms and water levels

18) In Figure 4, the Y axis lecture?

19) Line 320-323, The authors mentioned acidification would result from nitrification. For the quartz-sand growth media, I cannot understand the nitrification and its following acidification in the media.

20) The whole text needs to be refined by removing some non-important contents (e.g., lines 560-567; lines 118-125). The authors should pay more attention to refine the introduction and discussion sections.

Comments on the Quality of English Language

The whole text needs to be refined by removing some non-important contents (e.g., lines 560-567; lines 118-125). The authors should pay more attention to refine the introduction and discussion sections.

Author Response

reply to Reviewer 2

1) Line 10, changed into "N uptake by the roots is dominated by..."  
R/ Done

2) Line 14, rations, changed into ratios
R/ Done

3) Line 20, changed into "has shown..."
R/Done

4) Lines 22-25, this sentence is not clear. Please rephrase it.
R/ Done

5) Lines 36-37: what is the meaning of 90 million of 60 kg coffee bags?
R/ The coffee market and trade is expressed in millions of coffee bags, not in metric tons,  each coffee bag has the weight of 60 kg.  

6) Line 44: The worldwide coffer production it is under threat..., changed into The worldwide cof-fer production is under threat.... In the whole text, some similar expressions are available.
R/ Done

7) Line 49, what is the meaning of -S?
R/ S means climate change scenarios, defined by the po-tential or projects mean air temperature increase.

8) Line 73: Mineral fertilization it is an important driver of the productivity ..., the same as 6). Please delete the word "it".
R/ Done.

9) Lines 74-78 The sentence needs to be refined.
R/ Done.

10) Lines 83-84, what is the meaning of the degree of fertilizer used?
R/ Degree means how much N the fertilizer have per 100 kg of Fertilizer For Example urea: 46% N-0-0, or DAP 18% N and 46 P2O the degree is 18-46-0 etc. The line was rephrased.  

11) Lines 100-101, the sentence is not clear.
R/ The line was rephrased.  

12) Lines 118-126. this paragraph may be deleted. Anyway this introduction needs to be refined by removing unnecessary contents.
R/ Adjusted.

13) In Table 1, the word P-value can be removed. At present significant differences between treatments?
R/ Adjusted.

14) In Figure 1, "...soil with water stress (B)", changed into "...soil with water stress (D)".
R/ Ad-justed.

15) Line 227, ... using organic soil as a growth medium." However, in the other places, the au-thors used soil as a growth medium.
R/ Adjusted.

16) Table 3, in the soil, changed into growth medium. The descriptions for significant differences are not clear for capital and small letters. The same for Figures 4 and 5 and other tables or Figures. Also, significant differences in Table 4 are indeed not unknown.
R/ Adjusted the title. The differences between uppercase and lower are related with the statistical analysis, sepa-rated variance analysis was done for NH4-N and NO3-N content in the growing medium per treatment (Quartz-sand and Soil). Figure 5 was changed, because every years was analyzed independently.

17) 2.4 section, nitrogen forms and water levels.
R/ Adjusted.

18) In Figure 4, the Y axis lecture?
R/ Adjusted.

19) Line 320-323, The authors mentioned acidification would result from nitrification. For the quartz-sand growth media, I cannot understand the nitrification and its following acidification in the media.
R/ Nitrification is the transformation of the Ammonium into Nitrates and in that process 4H+ are release increasing the acidification of the growing medium 2NH4+ + 3O2 = 2NO2- + 4H+ + 2H2O.

20) The whole text needs to be refined by removing some non-important contents (e.g., lines 560-567; lines 118-125).
R/ Adjusted

21) The authors should pay more attention to refine the introduction and discussion sections.
R/ Done also as a recommendation of the reviewer 1.

Comments on the Quality of English Language
The whole text needs to be refined by removing some non-important contents (e.g., lines 560-567; lines 118-125). The authors should pay more attention to refine the introduction and discussion sec-tions.

Reviewer 3 Report

Comments and Suggestions for Authors

Dear Authors, please find detailed notes in the manuscript.

Kind regards

Reviewer

Author Response

  • Line 87. R/. Revision of the units, the SI for metric tones is “t”, for hectare “ha” and for kilogram “kg”.
  • Comments from the Table 1 R/ adjusted
  • Suggestion in the Table 2 R/ Changed Soil by soil.
  • Figure 2 R/ adjusted also following the suggestions from Reviewer 1.
  • Table 3 R/ We do not change the units, those are units also used in soil analysis and reports.
  • Figure 3, 4 and 5. R/ adjusted as well.
  • Line 290.
  • Line 299, Table 4. R/ The statistical analysis of the Chlorophyll content was done separately, one ANOVA for the data from February to October and other ANOVA only for the mean values.
  • Line 625. R/ Clarification of the harvest assessment is included in the lines 725 to 728.

Reviewer 4 Report

Comments and Suggestions for Authors

Comments and Suggestions for Authors

Title: Physiological and Agronomical Response of Coffee to Different Nitrogen
Forms with and without Water Stress

Dear Authors

 The subject is very interesting and correspond to the Plants journal’s profile. The presented manuscript addresses current problems regarding nitrogen fertilization of coffee crops in diverse water conditions. General value of the MS is good.

The aim of conducted study was compare the influence of several nitrogen forms NH4+ and NO3- and combinations in growth, nutrient uptake, photosynthesis, chlorophyll content and soil fertility using quartz-sand and organic soil as a growing medium in coffee production systems.

General remarks

In order to increase the usefulness of the article, Authors must refer to the following points. Additions should be made to increase the scientific value of the manuscript.

1.      Results: subsection 2.2.  and Table 2 - macronutrient uptake should be given in

g. plant-1.

2.      Discussion: You should once again verify the discussion of the results and remove those fragments that are not thematically related. Sequences that are loosely related to the presented results should be removed.

3.      Materials and Methods: subsection 4.1. - What was the purpose of testing on quartz sand? Reaction reagents for the Mehlich 3 method should be provided. CaCl2 concentration should be added. SI units should be used. Line 682 - More details regarding the determination of organic carbon should be added. Lines 742-743 The record should be standardized: chemical formulas or full names of compounds.

4.      Conclusions: A statement should be added: Is there a need for further research?

Specific comments

Line 26 – it should be: ammonium form (NH4+): nitrate form (NO3-). The ionic forms of nitrogen must be written correctly throughout the MS.

Reference no. 32 not cited in the text of the manuscript.

Lines 630 and 708 should be: 4.1. and 4.2. respectively.

Lines 650-654 The purpose of the research should be included at the end of the Introduction.

Best regards

Author Response

reply to Reviewer 4.

In order to increase the usefulness of the article, Authors must refer to the following points. Additions should be made to increase the scientific value of the manuscript.

1.      Results: subsection 2.2.  and Table 2 - macronutrient uptake should be given in g. plant-1.
R/ Adjusted

2.      Discussion: You should once again verify the discussion of the results and remove those fragments that are not thematically related. Sequences that are loosely related to the presented results should be removed.
R/ Adjusted according with the reviewer 1 as well.

3.a      Materials and Methods: subsection 4.1. - What was the purpose of testing on quartz sand?
R/ The objective to test quartz-sand was to avoid the interaction of the soil nutrient and buffer capacity on the N nutrition, and test the performance of the N forms without organic N interferences.

3.b. Reaction reagents for the Mehlich 3 method should be provided.  
R/ Included

3.c. CaCl2 concentration should be added. SI units should be used.
R/ Included

3.d.  Line 682 - More details regarding the determination of organic carbon should be added.
R/ Old and well know method to stimate soil organc carbon, avoiding the use of Potassium Dichromate.

3.3. Lines 742-743 The record should be standardized: chemical formulas or full names of compounds.
R/ Included

4.      Conclusions: A statement should be added: Is there a need for further research?
R/ Included

Specific comments
Line 26 – it should be: ammonium form (NH4+): nitrate form (NO3-). The ionic forms of nitrogen must be written correctly throughout the MS.
R/ Included

Reference no. 32 not cited in the text of the manuscript.
R/ Included

Lines 630 and 708 should be: 4.1. and 4.2. respectively.
R/ Adjusted

Lines 650-654 The purpose of the research should be included at the end of the Introduction.
R/ The main objectives of the research are describe at the end of the introduction, the reason of describe the aim of the greenhouse trials in this session is to try to provide a common thread to the methodology, and differentiate the main propose of  each one